# Dynamics of the Transcriptome Response to Heat in the Moss, *Physcomitrella patens*

**DOI:** 10.3390/ijms21041512

**Published:** 2020-02-22

**Authors:** Osama Elzanati, Said Mouzeyar, Jane Roche

**Affiliations:** Université Clermont Auvergne, INRAE, GDEC, Campus Universitaire des Cézeaux, 1 Impasse Amélie Murat, TSA 60026, CEDEX 63178 Aubiere, France; Osama.Elzanati@uca.fr (O.E.); Said.Mouzeyar@uca.fr (S.M.)

**Keywords:** *Physcomitrella patens*, heat stress, transcriptome, RNASeq

## Abstract

Thermal stress negatively impacts crop yields, and as the overall temperature of the earth’s atmosphere is gradually increasing, the identification of the temperature transduction pathway of the heat signal is essential in developing new strategies in order to adapt plant breeding to warmer climates. Heat stress damages the molecular structures and physiological processes in plants in proportion to the level and duration of the stress, which leads to different types of responses. In general, plants respond more efficiently when they are first subjected to a moderate temperature increase before being subjected to a higher temperature stress. This adaptive response is called the acclimation period and has been investigated in several plant species. However, there is a lack of information on the dynamic of the Heat Shock Response (HSR) over a continuous period of temperature rise without an acclimation period. In this paper, we investigated the effects of mild (30 °C) and high (37 °C) continuous heat stress over a 24-h period. Through RNA-Seq analysis, we assessed the remodeling of the transcriptome in the moss *Physcomitrella patens*. Our results showed that the 30 °C treatment particularly affected the expression of a few genes at 1 and 24 h, suggesting a biphasic response. Up-regulated genes at 1 h encode mainly HSR proteins (protein folding and endoplasmic reticulum stress), indicating an early heat response; while the up-regulated genes at 24 h belong to the thiamine biosynthesis pathway. In contrast, the genes involved in photosynthesis and carbon partitioning were repressed by this treatment. Under a higher temperature stress (37 °C), the induction of the HSR occurred rapidly (1 h) and was then attenuated throughout the time points investigated. A network approach (Weighted Gene Correlation Network Analysis, WGCNA) was used to identify the groups of genes expressing similar profiles, highlighting a HsfA1E binding motif within the promoters of some unrelated genes which displayed rapid and transient heat-activation. Therefore, it could be suggested that these genes could be direct targets of activation by a HsfA1E transcription factors.

## 1. Introduction

Past and recent reports indicate that the planet earth is experiencing a period of climate change with global warming that is giving rise to frequent episodes of extreme and violent weather events [1]. As sessile organisms, plants have developed appropriate strategies to cope with the fluctuating weather on a daily basis and on a longer timescale as well. Indeed, extreme temperatures have been shown to negatively impact the plant growth and production in cereals, such as in rice plants submitted to 9 °C above the optimal temperature [2] or in wheat plants to 8 °C [3]. In addition, disruption of fundamental processes, such as photosynthesis and carbon uptake, can lead to defects in vegetative development such as fewer, smaller and/or malformed organs [4] leading to flower abortion [5] and a large reduction in yield [6]. However, the negative effect of heat stress on plants is dependent on several parameters, such as the intensity, duration and amplitude of temperature variations [7,8].

Within a short time under heat stress, increases in cytosolic calcium concentrations activate Calmodulin (CaM) proteins, which then activate Heat Shock Transcription Factors (HSFs). Activated HSFs bind to promoters and initiate the transcription of heat responsive genes, such as those encoding Heat Shock Proteins (HSPs). Plants that develop this rapid Heat Stress Response (HSR) could be more thermotolerant to subsequent episodes of heat stress. Therefore, this process is responsible for the acclimation phenotype observed in plants [9]. However, the HSR is a complex response needing the coordinated action of hundreds of genes [10].

The moss *Physcomitrella patens* (*P. patens*) is used as a model organism to decipher the adaptation mechanisms deployed by plants to conquer and expand in different geographical regions [11] and compelling genomic and cellular resources, describing this phenomenon, are publicly available [12]. In particular, this model is used to uncover the physiological and molecular responses of plants to abiotic stress such as salinity, drought or heat [13,14,15,16]. It has been shown that an increase in temperature modifies the calcium influx through changes in membrane fluidity [17]. When *P. patens* protonemata are first grown for a short period at 32 °C (i.e., acclimation period), it has been demonstrated that the plants were more tolerant to a subsequent stress at 35 °C and displayed a weaker HSR than did non-acclimated plants [18,19]. The authors also demonstrated that the difference in temperature change is more determinant than the absolute temperature. The heat shock response in moss plants is regulated by specific calcium-permeable channels in the plasma membrane [18]. Although the moss *P. patens*, a land plant model, and *A.thaliana* diverged 420 million years ago, it has been shown that a cyclic nucleotide-gated calcium channel (CNGC), called CNGCb, is one of the primary heat sensors in both organisms [20], demonstrating that some basic steps in the HSR have probably been conserved between moss and terrestrial plants. Besides the HSR basic mechanisms, it has been shown that alternative mRNA splicing is involved in the regulation of the HSR in *P. patens* [21]. In particular, these authors suggested that the alternative splicing of mRNA encoding key genes, such as heat shock factor *HsfA2*, could ensure a fine tuning of the HSR.

While *P. patens* is routinely used as a plant model in order to understand the acclimation phenomenon (i.e., a short treatment with a non-damaging temperature before a second treatment with a higher temperature), there is a lack of information on this organism concerning the dynamics of the HSR over a longer period of temperature rise without acclimation. In this study, we analyzed the transcriptome dynamics of *P. patens* protonemata grown continuously at 30 °C or continuously at 37 °C compared to being grown at a standard temperature (24 °C), for 4 time points over 24 h (1, 6, 12, 24 h).

## 2. Results

After mapping sequencing reads, raw data were standardized and filtered using the criteria described in the Materials and Methods section. Among the 32,926 loci containing protein-coding CDS in *P. patens* genome; weakly expressed genes were filtered out using the cpm function in edgeR and only genes with a CPM value >0.5 in at least two samples were retained. These steps left 21,429 genes which were tagged as expressed and used for further analysis.

### 2.1. Global Analyses

Expression data standardized by the voom method [22] were used in a Principal Component Analysis (PCA). Figure 1 shows the distribution of samples according to the treatment. It can be noticed that biological replicates (Rep1 and Rep2) of each treatment are clustered and samples of the same treatment are distributed along the first axis of the PCA, which captures a variability corresponding here to 38%. Along the first axis, one group corresponds to the treatment at 24 °C, one is the intermediate group corresponding to 30 °C and the third one regroups samples treated at 37 °C. To contrast, the samples are distributed by the time point of treatment along axis 2, which represents 17% of the variability. Interestingly, only the 1-h time point at 37 °C is clearly distinguishable from the remaining 37 °C time points. In this respect, it can be noticed that the 1-h and 24-h samples at 24 °C are clustered, while the corresponding samples at 30 °C and 37 °C are more dispersed along the second axis. In this way, heat-treated samples and controls have been compared considering each time point. We also conducted a clustering analysis (distance: Euclidean, average) that shows three large groups: one group contains the samples treated at 24 °C and 30 °C, one group contains only samples treated at 37 °C for 1 h and, finally, a third group comprises the other samples (6, 12 and 24 h) of the treatment at 37 °C (Figure 1). It is worth note that a large part of the variability could be explained by an early response to heat stress (1 h) while the remaining one depends on the level of the heat stress (37 °C vs. the others).

### 2.2. Identification of Differentially Expressed Genes (DEGs) and the Ontology of the DEG

In our experimental design, four time points were realized (1 h, 6 h, 12 h and 24 h) for each treatment (24 °C (control), 30 °C and 37 °C) in order to compare samples by pairs at each time point. The DEGs were then classified as either up-regulated or down-regulated in comparison with the 24 °C control. To gain an overview of the temperature-affected metabolic pathways in the samples, the annotation of the DEGs, using Gene Ontology (GO) terms, was compared with the whole genome annotation [23], using Fisher’s exact test to detect specific GO term enrichment.

In total, we identified 6373 unique DEGs (differentially expressed in one or more conditions) distributed as follows: 2960 were activated (“up-regulated”) (Figure 2, panel A) in one or more of the treated samples versus the control samples with 129 genes that were affected by both heat treatments; 3475 genes were repressed (“down-regulated”) (Figure 2, panel B), with 506 in common. Detailed information on the DEGs is given in Appendix A.

### 2.3. Mild Temperature Effect (30 °C Treatment)

Under 30 °C, only 30 genes were up-regulated at 1 h and were associated mainly with the GO terms related to protein folding, response to hydrogen peroxide, response to endoplasmic reticulum stress and response to heat (Figure 3). Then, interestingly, at 24 h, the only significant GO terms identified are related to thiamine biosynthetic process and response to vitamin B1 (Appendix A); and they represent the highest number of DEGs (186). Although no significant GO term enrichment was found at 6 h or 12 h, it is interesting to note that the DEGs at 6 h and 12 h are mainly involved in the heat response (genes encoding HSP proteins). Regarding the repressed genes, a high number of DEGs (618) were found at 24 h and are associated with photosynthesis and carbon fixation GO terms (Figure 4). Although no enrichment in any GO term was detected at 1 h, 6 h or 12 h due to the small number of genes differentially expressed, it has to be pointed out that the DEGs that were concerned are involved in carbohydrate metabolism (Appendix A).

### 2.4. High Temperature Effect (37 °C Treatment)

Concerning the 37 °C treatment, most of the DEGs were found to be differentially expressed at 1 h and 24 h. The GO terms of the genes expressed at 24 h were related to photorespiration process (glyoxylate cycle) and gene silencing (559); and at 1 h, to protein folding (479). Genes expressed continuously from 1 h and maintained during the treatment up to 24 h showed GO terms related to response to heat, response to endoplasmic reticulum stress and response to hydrogen peroxide (Figure 5). Regarding the down-regulated genes, most of the DEGs are expressed after 12 h (377) and 24 h (854) of treatment and are associated with oxazole and thiazole metabolic processes; and photosynthesis, respectively (Figure 6). Actually, early after heat treatment (6 h), it seemed that the defense mechanisms, as well as the negative regulation of cell cycle, were repressed (Appendix A). In the later phases (12 h and 24 h), the negative effects on photosynthesis and carbon fixation became very significant.

To gain insight into the dynamics of gene expression at 37 °C over the period tested, we aimed at identifying the genes that displayed similar expression profiles (co-expressed genes). For this purpose, we used the weighted gene co-expression network analysis (WGCNA) to identify the modules of highly correlated genes resulting in 7844 DEGs. These were subsequently explored for enrichment in specific GO terms. Using this method, 6373 co-expressed DEGs were grouped into 7 color-coded modules (yellow = 563 genes, turquoise = 1624 genes, red = 162 genes, black = 118 genes, brown = 977 genes, blue = 1209 genes, green = 249 genes) (Appendix A). The remaining genes (i.e., 1471 genes) which displayed more erratic expression profiles and were grouped into the grey module.

In our WGCNA analysis, each module is summarized by its eigengene expression representing the highest variation in the expression profile of the considered module (1st Principal Component). Eigengene expressions of the seven modules containing highly correlated genes were plotted over 24 h, illustrating both the control and the heat-treated samples together. The yellow and turquoise modules contain genes whose expression is up-regulated by high temperature. The maximal induction was observed at 1 h after the treatment in the yellow module, while it starts mainly at 6 h with a maximum at 12 h after heat treatment in the turquoise module (Figure 7a,b). The black, blue, brown and green modules contain genes whose expression was down-regulated by temperature increase. The black module contained genes whose expression was very low at 1 h post-treatment (Figure 8e), the green module contained genes whose expression was low at 24 h (Figure 8a), while the blue (Figure 8b) and brown (Figure 8c) modules contained genes whose expression started to decrease at 6 h, with this diminution lasting up to 24 h after heat treatment. The last module (red module) contained genes whose expression profile was more complex with a marked reduction of expression at 6 and 12 h after heat treatment (Figure 8d). GO term enrichment analysis for each module (BIOLOGICAL_PROCESS) is presented in Appendix A. The results indicated that, the yellow module was enriched in 6 GO terms, including: response to heat (FDR = 2.95 × 10^−14^), response to hydrogen peroxide (FDR = 1.45 × 10^−11^), response to endoplasmic reticulum stress (FDR = 1.09x10^-8^), protein refolding (FDR = 6.1 × 10^−6^). The turquoise module was also enriched in 6 GO terms including: DNA repair (FDR = 0.0014), protein targeting to mitochondria (FDR = 0.007), DNA conformation change (FDR = 0.02323). The green module was enriched in 14 GO terms including: Cell wall organization (FDR = 0.0049), external encapsulating structure organization (FDR = 0.0049) and Plant-type cell wall assembly (FDR = 0.045). The brown module was enriched in 63 GO terms, including: isopentenyl diphosphate biosynthetic process (FDR = 1.37 × 10^−32^), photosystem II assembly (FDR = 2.3 × 10^−18^), Starch biosynthetic process (FDR = 6.6 × 10^−8^), Photorespiration (FDR = 0.000015) and Defense response to bacterium (FDR = 0.000019). The blue module was enriched in 18 GO terms, including microtubule-based movement (FDR = 0.018), negative regulation of metaphase/anaphase transition of cell cycle (FDR = 0.018) and mitotic nuclear division (FDR = 0.018). The red module was enriched in 2 GO terms related to response to light intensity (FDR = 0.004) and response to red or far-red light (FDR = 0.006). Finally, no enrichment in any GO term was found in the black module.

### 2.5. Conserved Motifs within Promoters of Correlated Genes

Following the identification of the expression modules containing genes whose expression is highly correlated, we investigated whether members of a given module share common motifs within their promoters. To test this hypothesis, the top 100 genes displaying the highest KMM (i.e., the highest module membership) were selected from each module and 1500 bp upstream of their putative ATG codon were scanned using MEME suite (http://meme-suite.org/tools/meme). The de novo search was limited to only 4 different motifs without any limitation on the number of repeated motifs per sequence. Motifs that were detected by MEME were then subjected to a homology search using the motif comparison tool Tomtom (Tomtom Hits) (see Section 4 Materials and Methods).

For each module, a substantial number of sequences were found to share the same motif. For instance, 16 sequences in the red module were found to contain the motif “TATATATATATA” in their promoter, while up to 75 sequences in the brown module were found to contain the motif “GGRGRGRGRGRG”. Detailed results of each module are provided in Appendix A. Table 1 presents an example of the motifs discovered in the yellow module.

As a proof-of-concept, we chose the yellow module because it contained a majority of genes known to be rapidly activated by high temperature, such as those coding for HSPs. In this context, it was expected to find heat stress elements (HSE) within the promoters of these genes. Finding HSE could be considered as a proxy of the relevance of our approach. The cis-motifs within the promoters of co-expressed genes present in the other modules are provided in Appendix A. Interestingly, among the top 100 genes of the yellow module, which was found enriched in GO terms related to heat stress, 68 sequences were found to contain the motif “AAAATAWAWAWA” which displayed similarities with the motif “C2C2dof_tnt.At4g38000_col_a_m1” in *A. thaliana* that is recognized by a DOF transcription factor (AT4G38000) (DNA binding with one finger 4.7). 70 sequences shared the motif “GGRAGAGRARGA” with similarities with “C2H2_tnt.TF3A_col_a_m1” recognized by *AT1G72050* which encodes the transcriptional factor TFIIIA required for transcription of 5S rRNA gene. 49 sequences contained the conserved motif “TGBTGKTGKTG” recognized by *AT1G18960*, a myb-like HTH transcriptional regulator. Finally, 49 out of the top 100 genes in this module shared, within their promoter, the conserved motif “GAAGYTCTAGA” which showed similarities (*e*-value = 0.000000039) with the Heat Shock Elements (HSE) recognized by the heat shock factors (HSF) in *A. thaliana*. The identification code of this element is HSF_tnt.HSFA1E_col_a_m1 in the *A. thaliana* DAP-seq database [24]. The set of 49 sequences (Appendix A) containing this motif comprised several genes annotated as Heat Shock related proteins such as *Pp3c20_14730* and *Pp3c17_2560* (DNAJ heat shock family protein), *Pp3c18_16670* and *Pp3c19_22640* (*HSP20-like*), *Pp3c14_19430* and *Pp3c11_1010* (*HSP70*). In addition to heat shock proteins, this set contained other genes, such as *Pp3c3_15530* and *Pp3c1_12850*, both encoding proteins with similarities with Bax inhibitor-1 as well as *Pp3c9_8740* and *Pp3c15_4650*, both encoding putatif RHOMBOID-like-protein-14. Similarly, *Pp3c7_700*, which encodes a putative RING/U-box protein, was found within this set of genes (Appendix A). This suggests a possible direct activation of these genes by Heat Shock Transcription Factors.

## 3. Discussion

### 3.1. Global Effects of Temperature on the P. patens Transcriptome

The special IPCC global warming report (https://www.ipcc.ch/sr15/) indicated that the increase in global mean surface temperature (GMST) by 1.5 °C is likely to increase the frequency and severity of extreme events, such as flooding, drought and heatwaves. In laboratories, *P. patens* is routinely cultivated at temperatures around 25 °C. (http://moss.nibb.ac.jp/). Therefore, the aim of this work was to assess the dynamics of the responses of *P. patens* to changes in continuous temperature over a short period of time and two temperatures were chosen (30 and 37 °C) as mild (warming) and strong stress inducers to be compared to the optimal (control) temperature of 24 °C. Multidimensional analysis of the RNASeq data via PCA and clustering indicated a clear separation between the samples belonging to 24, 30 and 37 °C treatments suggestive of different responses depending on the temperature intensity. However, all the 30 °C samples were grouped between the 24 °C samples and the 37 °C samples in the PCA plot, which could be explained by a gradual response of *P. patens* to temperature increase. Indeed, Claeys et al. [25] showed that *Arabidopsis thaliana* plants could adapt their response to abiotic stress intensity. Similarly, Kumar and Wigge [26] experimentally showed that at least one HSP70 member displayed a uniform linear expression induction upon constant temperature increases which supports the hypothesis that the modulation of expression depends, at least in part, on the heat stress intensity. It also has to be noted that time point samples from 1 to 24 h are spread along the second PCA axis. However, the magnitude of variation was higher between the 37 and 30 °C samples than between the 24 °C samples. In particular, the 1 and 24 h samples at 24 °C were close to each other while the corresponding samples at 30 and 37 °C were highly dissimilar. This result suggests that a part of the variability observed may involve the circadian rhythm and that this latter may be modified by a high temperature treatment. In their review, Gil and Park [27] suggested that HSPs are important to preserve the circadian clock under temperature stress. Moreover, although the circadian clock enables plants to maintain plant growth and vigor, some of the genes controlling it may be modified by increasing temperature. Blair et al. (2019) [28] also showed that some genes may be specifically expressed in response to heat depending on the time of day. In total in our experiments, 19 genes belonging to those central clock genes (extracted from Holm et al. 2010 [29]), such as *CCA1*, have been found to be differentially expressed under 37 °C and 30 °C at different times (Appendix A circadian genes). Those genes have also been described in the preservation of the circadian rhythm under cold treatment [30]. 

### 3.2. Dynamics of the Transcriptional Response of P. patens to High Temperature (37 °C)

The molecular and physiological responses of *P. patens* to severe heat stress have been the scope of many studies. For instance, membrane composition and fluidity [17] and extracellular calcium influx through specific cyclic nucleotide-gated channels (CNGCs) [20] constitute primary thermosensors capable of triggering heat-shock response (HSR). The knockout of a small heat-shock protein sHSP16.4 compromised the recovery process after heat stress application [16] demonstrating that the HSR has a protective role in *P. patens* cells. Saidi et al. [19] have tested the soybean promoter of the same small heat-shock protein (Gmhsp17.3B) to monitor its activation by heat stress, which reaches its maximum when protonemata were placed for 1 h at 38 °C. However, prolonging the duration of treatment up to 2 h did not result in further activation of this promoter. Saidi et al. [18] confirmed this finding and showed that the alternation of inducing and recovery temperatures resulted in more HSP induction than a continuous heat stress, suggesting that the heat stress response was attenuated and necessitated a recovery period to be reset and become fully responsive. In our study, 70 HSP-related genes were found up-regulated by 37 °C at 1 h, 44 genes at 6 h, 52 genes at 12 h and 44 genes at 24 h. Among these genes, 20 were exclusively found at 1 h whereas only 4 genes were found exclusively up-regulated at 24 h. This result indicated that the induction of HSP-related genes is rapidly followed by an attenuation phase. While the findings of Saidi et al. [18,19] were based on the induction of an HSP promoter; here, we extended this observation to several *HSP* genes, supporting the hypothesis that the attenuation phase could be a general process. Although the molecular mechanism of the attenuation process is still not fully known, some hypotheses suggest that this process could be mainly achieved through the inactivation of Heat Shock Factors by HSP70 chaperones which act directly on the Hsf1 transactivation domain and repress the transcription of heat shock genes [31]. Another possibility could be a negative regulation of HSR by a class B heat shock factor, such as Hsf2B in *A. thaliana* [32]. Indeed, 4 genes with homology to the *A. thaliana* Hsf2B gene (*AT4G11660*) were found in the genome of *P. patens*, among which, two were found up-regulated at 1 h at 37 °C: *Pp3c21_18820* (fold change =4.6, *p*-value=0.00000503) and *Pp3c22_9250* (fold change = 3.49, *p*-value = 0.000274674) suggesting a possible negative regulation exerted by these Hsf2B-homologous genes on the HSR. Alternatively, Li et al. [33] suggested that the activation of one of the Endoplasmic Reticulum Associated Degradation (ERAD) pathways (Cytosolic Protein Response (CPR)) by heat stress may lead to the termination of the HSR in an attempt to avoid excessive responses in *A. thaliana* plants and maintain protein homeostasis through Hsf2B. However, the involvement of Hsf2B-related genes as negative regulators or the ERAD pathways in inhibiting the HSR response need to be experimentally addressed in *P. patens*.

Besides the up-regulated genes, we found that the number of down-regulated ones steadily increased with the duration of heat stress. Those genes are mainly involved in protein folding, which may indicate that this process is directly affected by the heat. Additionally, Larkindale and Vierling [34] demonstrated that more attention should be paid to those genes whose expression is repressed by heat stress. In particular, they found that many of the repressed genes are involved in general growth metabolism but the genes related to Programmed Cell Death (PCD), such as disease resistance genes, were also found down-regulated in *A. thaliana*. Those authors suggested a fine balance between the activation of prosurvival genes and the repression of genes promoting cell death. Consistent with this statement, among the genes annotated as resistance genes, we found that 13 were down-regulated by 37 °C at 1 h, 20 genes at 6 h, 24 genes at 12 h and 25 genes at 24 h. This finding suggests that crosstalk between the activation of HSR and the repression of PCD could occur in order for the new metabolism statement to better adapt to the new temperature conditions. However, this assumption needs further investigation to be confirmed. As for the repression of HSR, it was reported that HsfB1 and HsfB2b are negative regulators of disease resistance in *Arabidopsis thaliana* [35]. Interestingly, conversely to mild temperature stress (30 °C), our results highlighted a down-regulation of the thiamine biosynthesis pathway at 12 h, which confirms the hypothesis that the mechanisms of thermotolerance induced in response to temperature may depend on the level of the temperature perceived.

### 3.3. Dynamics of the Transcriptional Response of P. patens to Mild Temperature (30 °C)

Only a very limited number of differentially expressed genes (i.e., 852 genes, see Appendix A) were identified using the limma method and protonemata subjected to a 30 °C treatment. In addition, the majority of DEGs were found at 1 h (53 genes) and 24 h (820 genes) after temperature treatment corresponding to a “biphasic” response with an early response at 1 h and a late response at 24 h. Enrichment in GO terms indicated that early response consisted of response to heat stress (HSR) such as “protein folding” and “endoplasmic reticulum stress” (ERS) whereas no enrichment was observed in the later stages (6 h, 12 h and 24 h after treatment). A careful examination of the DEGs indicated that 30 out of 49 up-regulated genes at 1 h belonged to these two classes; 1 out of 1 gene at 6 h, 6 out of 8 at 12 h and 10 out of 153 up-regulated genes at 24 h. This result suggested that 30 °C is at once perceived as a harmful temperature which causes damage to cellular proteins, hence activating cytosolic and endoplasmic reticulum chaperones in order to repair unfolded and misfolded proteins and to restore protein homeostasis [36]. This rapid response seems sufficient to alleviate the stressful treatment at 30 °C. The late response corresponded to a down-regulation of genes related to photosynthesis and carbon fixation suggesting that even though the stress response was limited in time, a long-term negative impact of a temperature rise could still be observed. It therefore seems that very soon after the application of the temperature of 30 °C (i.e., from 1 h), there is a response of the “stress response” type involving HSP-like genes, then there would be an acclimation phase which is ensued at 24 h by the activation of thiamine synthesis. Conversely, the negative effects illustrated by the repression of photosynthesis seem to occur only after 24 h. These observations also indicate that *Physcomitrella patens protonema* cells may respond to a temperature rise in a dose-dependent manner, which is consistent with the findings of Kumar and Wigge [26], who showed that the induction of a HSP70 gene in *A. thaliana* plants is proportionate to the ambient temperature rise between 12 °C and 27 °C, and that the negative effects may be proportionate to the deviation from the optimum. The effects of similar mild temperatures (increase of ambient temperature by 8 °C above the optimal growth temperature) have been tested on bread wheat (*Triticum aestivum*, SxB139 and SxB49 recombinant inbred lines) and yield was reduced by approximately 17% [3] whereas a reduction of up to 50% was observed at high temperatures [37,38]. 

GO enrichment analysis also indicated that, 24 h after the application of the 30 °C treatment, there is an induction of genes related to thiamine biosynthesis. Thiamine (vitamin B1) was reported to participate both in normal growth and in the adaptation of plants to abiotic and biotic stresses. For instance, thiamine supply has been shown to mitigate the negative effects of salinity on growth in sunflowers [39]. Also, heat stress induced an accumulation of the thiamine biosynthetic enzyme hydroxyethyl thiazole phosphate synthase in *Populus euphratica* [40]. Thus, an up-regulation of genes related to the thiamine biosynthesis in *P. patens* at 30 °C may suggest that this compound participates in the acclimation process. Furthermore, Rapala-Kozik et al. [41] showed an increase of total thiamine content and a gain in the thiamine activating enzyme, thiamine pyrophosphokinase, in maize seedlings submitted to high salt and drought stresses. The authors attributed this increase to the accumulation of Reactive Oxygen Species (ROS) in response to abiotic stresses. Alternatively, the activation of this pathway could be explained by the need to compensate the high demand on thiamine which is used as a cofactor in numerous cellular processes, in particular, for the synthesis of stress-related proteins and stress-related compounds. However, it would be interesting to test, experimentally, if supplying a *P. patens protonema* culture with exogenous thiamine would alleviate the effects of a moderate heat stress.

### 3.4. Attempt to Identify de Novo Candidate Players in Heat Stress Response

Larkindale and Vierling identified the sets of genes in *A. thaliana* involved in response during acclimation to heat stress [34]. They classified the genes responding to heat in 73 clusters, some of which were enriched in genes encoding HSPs. The promoters (i.e., 1000 bp) of the genes that belonged to these particular clusters contained the palindromic motif GAAnnTTC which binds heat shock factors Hsf1AE [42]. Furthermore, these clusters contained other genes (some with unknown functions) than HSPs. In our work, we used a network approach (WGCNA) to identify the groups of genes with similar expression profiles. We then made the assumption that genes belonging to the same module (cluster) are likely to have a similar repertoire of cis-regulatory elements within their respective promoters. Indeed, the promoters of genes in the yellow module, which displayed rapid and transient heat-activation, were enriched with the HsfA1E motif. Among the top 100 genes of this module, 49 genes contained a putative heat stress element. For instance, *Pp3c9_6640*, which encodes a protein similar to heat shock protein 81.4 in *A. thaliana*, contained one canonical HsfA1E-binding site GAAnnTTC. Similarly, *Pp3c13_19370*, annotated as an E3-Ubiquitin protein ligase (Ring/Ubox family), contained the same motif. Interestingly, the *Pp3c13_1670* gene, with similarities to a myeloid leukemia factor, contained 10 HsfA1E motifs. However, while these observations suggest that these genes could be direct targets of the activation by the HsfA1E transcription factor, this hypothesis should be verified using appropriate experiments. Indeed, Maruyama et al. [43] showed that the heat stress element (HSE) should be included in a particular 18 nucleotide sequence to yield a heat stress responsiveness in some crop plants. Nevertheless, this approach provides a set of new candidate genes amenable to experimental verification as to whether they are direct targets of HSF transcription factor as well as the classical HSPs encoding genes.

## 4. Materials and Methods

### 4.1. Plant Material

Protonemata of the *Physcomitrella patens* Gransden strain (The international Moss Stock Center, Germany, # 40001) were cultivated axenically in Petri dishes, containing a PpNH4 medium, as described by Ashton and Cove [44], supplemented with 7 g·L^−1^ of agar, in a growth chamber maintained at 24 °C under a photoperiod of 16/8 h and an irradiance of 100 µmol·m^−2^·s^−1^.

### 4.2. Heat Treatments

*P. patens* protonemal tissue was treated at 2 temperature levels (37 and 30 °C) at four time points (1, 6, 12 and 24 h). Two biological replications were realized. The temperature of the control treatment was 24 °C.

### 4.3. RNA Extraction

Total moss RNA was extracted from 100 mg of *P. patens* 7-day protonemal tissue from the heat and control treatments using Nucleospin^®^ RNA plant (Macherey-Nagel-Germany), DNase I treated with the kit following the manufacturer’s instructions for RNA-Seq analyses. Total extracted RNA was quantified using a spectrophotometer (nanodrop^®^) and the integrity was verified on 2% agarose gel.

### 4.4. Sequencing

Sequencing was carried out by Eurofin (Germany) using one lane and Illumina HiSeq 2500 using a 2 × 100 bp paired-end strategy. Briefly, mRNAs were purified using the polyA tract then used to construct strand-specific libraries. Each sample was tagged using 2 × 8 bp and after sequencing, the sequences were demultiplexed according to the 2 × 8 bp index code with 1 mismatch allowed. Libraries were generated using an Illumina Truseq Stranded mRNA kit following the manufacturer’s instructions (Kit version HiSeq SBS Kit v4, Illumina Way, San Diego, CA, USA). Library quality control and quantification were performed using Agilent Bioanalyser 2100 with High Sensitivity DNA Assay and qPCR for concentration determination before sequencing. The average insert size is approximately 340 bp (other specifications: Software: HiSeq Control Software 2.2.58, Illumina Way, San Diego, CA, USA). Sequences were deposited in the NCBI Sequence Read Archive (SRA) under the accession number GSE138467.

### 4.5. Mapping of Reads to P. patens Genome

Mapping of reads to reference sequences was performed using BWA-MEM and Galaxy platform (toolshed.g2.bx.psu.edu/repos/devteam/bwa/bwa_mem/0.7.17.1) [45]. The parameters were set as follows: “Algorithm for constructing the BWT index” = Auto. Let BWA decide the best algorithm to use. “Single or Paired-end reads” = Paired, “mean, insert lengths” = 100; “analysis mode” = Illumina.

The *P. patens* genome version Ppatens_318_v3.3, organized in 378 scaffolds and 32,926 loci containing protein-coding transcripts, was downloaded from (https://phytozome.jgi.doe.gov/pz/portal.html#!info?alias=Org_Ppatens:PhytozomeV12:Ppatens_318_v3.fa.gz) and used as the reference genome for mapping. In total, twenty-four BAM files, corresponding to 24 samples, were generated-2 biological replications x 3 heat treatments (24, 30, 37 °C × 4 time points (1, 6, 12, 24 h).

Raw read counts were created using *featureCounts* on a galaxy web server (Galaxy Tool Version: 1.4.6.p5 with default parameters [46]. The annotation version 3.3 of *P. patens* genome (file Ppatens_318_v3.3.gene.gff3, downloaded from https://phytozome.jgi.doe.gov/pz/portal.html#!bulk?org=Org_Ppaten), was used to estimate the raw expression of each gene (feature type filter = gene).

### 4.6. Data Transformation, Filtering and Normalization

All the steps of data normalization and the identification of differentially expressed genes were conducted using the limma package version 4.40.2 according to the procedure described by Law et al. [47]. Raw counts were converted to counts per million (CPM) using the cpm function in edgeR. Only genes with a CPM value > 0.5 in at least two samples were retained for further analysis. The genes that did not meet this criterion were considered as not expressed and thus filtered out. Expression of retained genes was normalized using Trimmed Mean of M-values (TMM) using the calcNormFactors function in edgeR [21]. This step yields a scaling factor that accounts for the library size. Finally, heteroscedasticity was removed using the voom method and library sizing. This normalized data was then used for global gene expression analysis (PCA and WGCNA, see below) and the detection of differential gene expression (DEG, see below).

### 4.7. PCA and Clustering Analysis

PCA analysis of the normalized data was conducted using FactoMineR package version 1.42. Hierarchical clustering was carried out using hclust in R with the euclidean distance and average method for agglomerations.

### 4.8. Identification of Differentially Expressed Genes (DEG)

To identify differentially expressed genes, eBayes and treat functions in the limma package were used to compare pairs of treated (30 °C or 37 °C versus control (24 °C) samples at 1 h, 6 h, 12 h and 24 h), yielding a total of 12 comparisons. A gene was considered as differentially expressed (DEG) if the absolute value of the log2 Fold Change (treated vs. control) was superior or equal to 1 (|log2FC| ≥ 1) with an adjusted *p*-value < 0.05. Up-regulated and down-regulated genes were identified if the value of log2FC was positive and negative, respectively.

#### 4.8.1. Identification of Co-Expressed Genes

A network of co-expressed genes was constructed using the Weighted Gene Co-expression Network Analysis (WGCNA) package in R [48]. Voom-normalized expression data, corresponding to the DEGs, were used to construct the network of co-expressed genes with the following parameters: A soft thresholding power of 18, networkType = “signed”, minModuleSize = 60, maxBlockSize = 2 000, corType = “pearson”, deepSplit = 2. CutHeight = 0.9. Initial modules with eigengene correlation ≥ 0.75 were merged. Non-assigned genes were grouped in the “grey” module. The modules of co-expressed genes were then color-coded.

#### 4.8.2. Annotation of the Set of DEGs

Although an annotation of *P. patens* genome was recently released [11], we used Blast2GO to reannotate the set of DEGs with *A. thaliana* proteome as reference. BlastP parameters were set at min. *e*-Value = 1 × 10^−4^, max Hits =20.

#### 4.8.3. GO Term Enrichment Analysis of the DEG and WGCNA Modules

Significant enrichments in GO terms in the DEG and WGCNA modules were identified using Fisher’s exact test with *P. patens* annotation version 3.3 as background [11] and a *p*-value threshold of 0.05 (Hochberg (FDR) as the multi-test adjustment method). 

#### 4.8.4. Identification of Conserved Motifs within the Promoters of DEG

The WGCNA identified sets of co-expressed genes which were arranged in color-coded modules. Each module could be represented by its “Module Eigengene” (ME) which corresponds to the first component of the PCA and which summarizes the expression of that module. Thereafter, genes within each module were ranked according to the correlation between their expression profile and the “Module Eigengene”. The top 100 genes in each module (i.e., displaying the highest correlation with the ME) were selected and 1500 bp upstream of their ATG initiation codon (promoter region) were retrieved from http://plants.ensembl.org.

Sequences were then scanned for the presence of conserved motifs using MEME Suite [49] with the following parameters: number of motifs: 4; minimum window size: 6; maximum window size: 12; markov_order: 0. The motifs that were detected by MEME were then subjected to a homology search using the motif comparison tool Tomtom version 5.O.5 [50] and Arabidopsis DAP (DNA Affinity Purification) motifs [24] as reference for comparison.

## 5. Conclusions

The effect of continuous high and mild temperatures over a relatively long period on the transcriptome of the moss *P. patens* was investigated for the first time. In this paper, we showed that *P. patens* responded differently according to the intensity of the applied temperature. A high temperature of 37 °C rapidly (1 h) induced the expression of a very large number of genes related to the Heat Shock Response (HSR) whereas, a very large number of genes were repressed after 24 h of treatment. Concerning the milder treatment (30 °C), the number of affected genes is relatively low and the most striking response occurs only 24 h after treatment with the activation of genes related to the thiamine synthesis pathway. Overall, a small percentage of differentially expressed genes (~10%) are common between the 30 °C and 37 °C treatments, suggesting different response signatures between these two temperatures and probably different adaptation mechanisms.

## Figures and Tables

**Figure 1 ijms-21-01512-f001:**
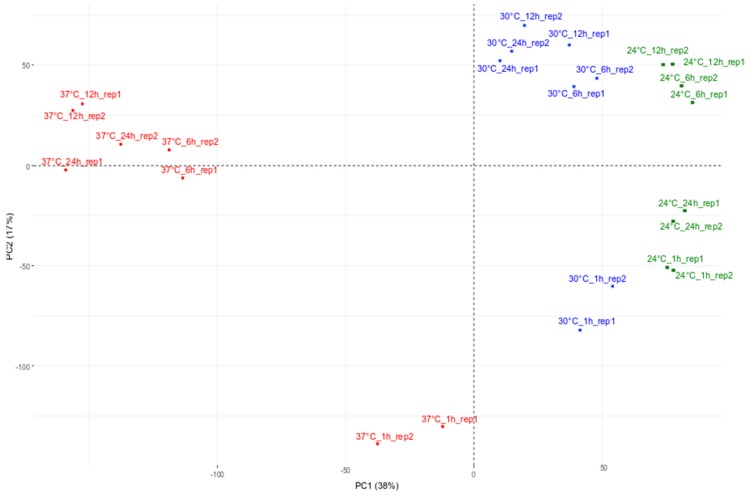
Plot showing the first two components of the principal component analysis (PC1 and PC2) representing the transcriptome of *P. patens* plants grown under heat treatments (30 °C in blue and 37 °C in red) and control conditions (24 °C in green) at 1, 6, 12 and 24 h. As an example: “24 °C_1h_rep1” corresponds to the first replicate of the treatment at 24 °C after 1 h.

**Figure 2 ijms-21-01512-f002:**
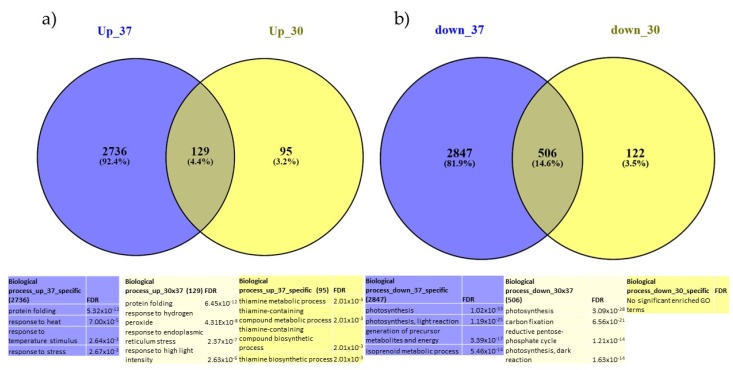
Venn diagram of expression data analysis at 37 °C and 30 °C of the (**a**) up-regulated genes and (**b**) down-regulated genes; and associated GO terms. Up_30 = DEG up-regulated at 30 °C at all time points, down_30 = DEG down-regulated at 30 °C at all time points. Up_37 = DEG up-regulated at 37 °C at all time points, down_37 = DEG down-regulated at 37 °C at all time points.

**Figure 3 ijms-21-01512-f003:**
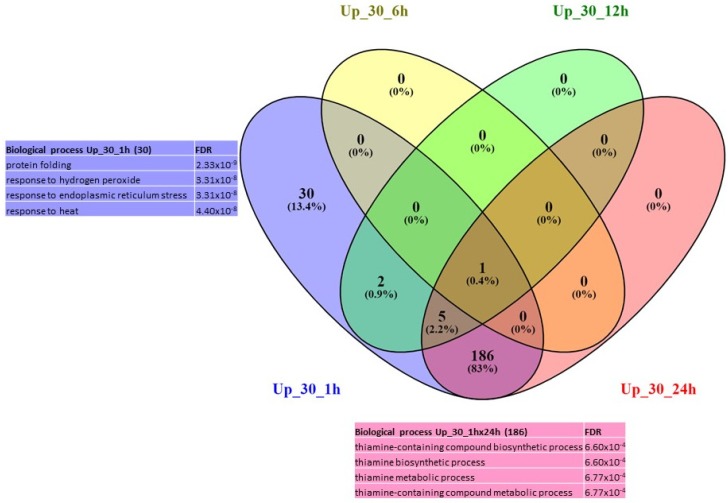
Venn diagram of expression data analysis at 30 °C of the up-regulated genes at 1 h, 6 h, 12 h and 24 h; and associated GO terms. Up_30_1h = up-regulated DEG at 30 °C at 1 h, Up_30_6h = up-regulated DEG at 30 °C at 6 h, Up_30_12h = up-regulated DEG at 30 °C at 12 h, Up_30_24h = up-regulated DEG at 30 °C at 24 h.

**Figure 4 ijms-21-01512-f004:**
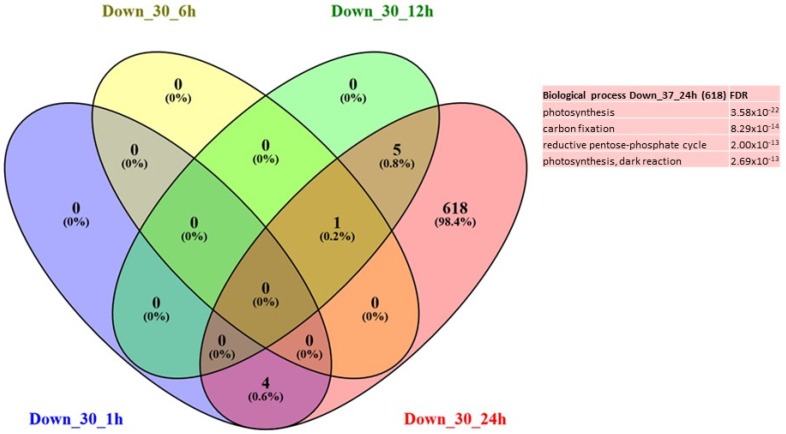
Venn diagram of expression data analysis at 30 °C of the down-regulated genes at 1 h, 6 h, 12 h and 24 h; and associated GO terms. Down_30_1h = down-regulated DEG at 30 °C at 1 h, Down _30_6h = down-regulated DEG at 30 °C at 6 h, Down _30_12h = down-regulated DEG at 37 °C at 12h, Down _30_24h = down-regulated DEG at 30 °C at 24 h.

**Figure 5 ijms-21-01512-f005:**
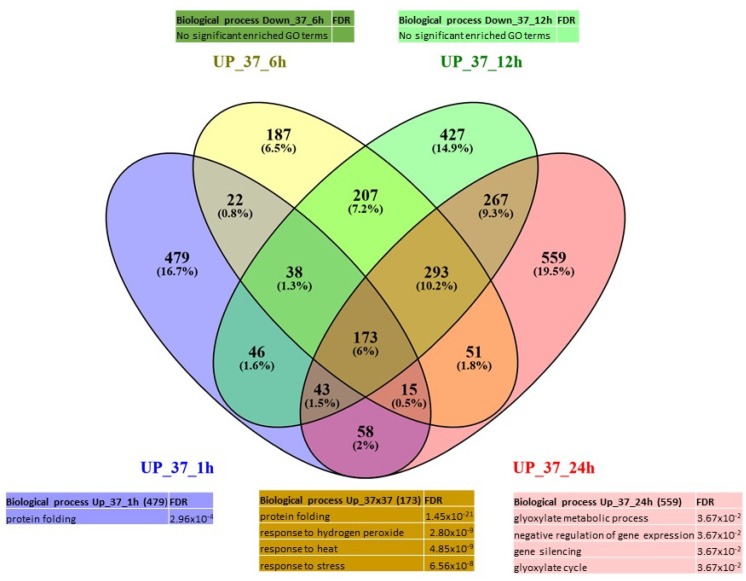
Venn diagram of expression data analysis at 37 °C of the up-regulated genes at 1, 6, 12 and 24 h and associated GO terms. UP_37_1h = up-regulated DEG at 37 °C at 1 h, UP_37_6h = up-regulated DEG at 37 °C at 6 h, UP_37_12h = up-regulated DEG at 37 °C at 12 h, UP_37_24h = up-regulated DEG at 37 °C at 24 h.

**Figure 6 ijms-21-01512-f006:**
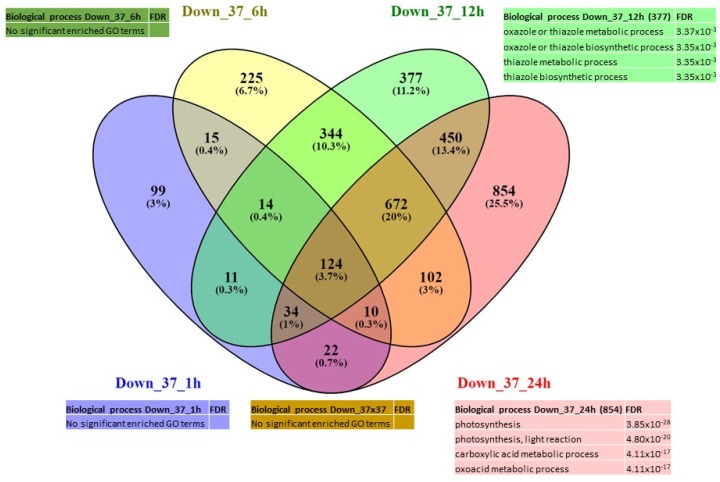
Venn diagram of expression data analysis at 37 °C of the down-regulated genes at 1, 6, 12 and 24 h; and associated GO terms. Down_37_1h = down-regulated DEG at 37 °C at 1 h, Down_37_6h = down-regulated DEG at 37 °C at 6 h, Down_37_12h = down-regulated DEG at 37 °C at 12 h, Down_37_24h = down-regulated DEG at 37 °C at 24 h.

**Figure 7 ijms-21-01512-f007:**
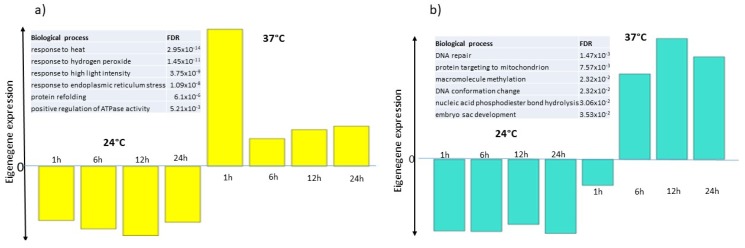
Eigengene expression of the yellow (**a**) and turquoise (**b**) WGCNA modules containing genes up-regulated by temperature increase. The histogram bars represent the eigengene expression of each module in samples grown at 24 °C or 37 °C for 1, 6, 12, 24 h. The GO term enrichment is represented in the left upper part of each figure panel.

**Figure 8 ijms-21-01512-f008:**
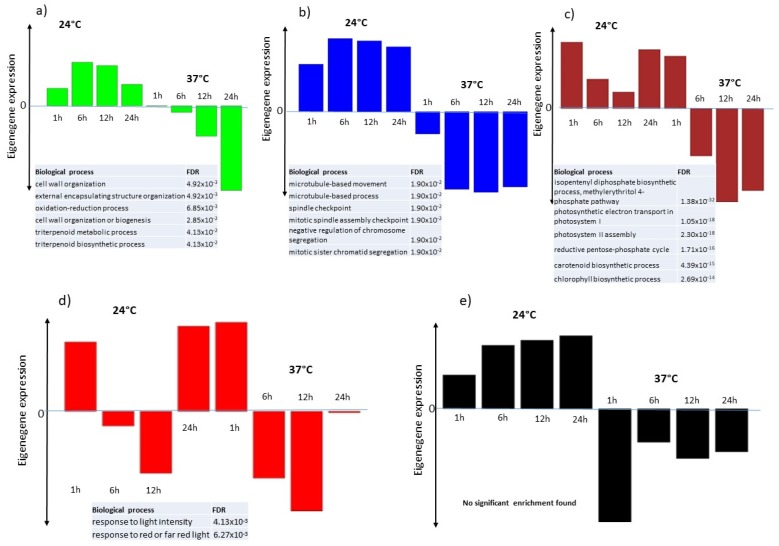
Eigengene expression of the green (**a**), blue (**b**), brown (**c**), red (**d**) and black (**e**) WGCNA modules containing genes down-regulated by temperature increase. The histogram bars represent the eigengene expression of each module in samples grown at 24 or 37 °C for 1, 6, 12, 24 h. The GO term enrichment is represented in the lower part of each figure panel.

**Table 1 ijms-21-01512-t001:** MEME Suite discovered motifs within 1500 bp upstream from the initiation codon and Tomtom Hits of the top 100 genes in the yellow module form the WGCNA.

	MEME_Discovered_Motifs	Tomtom_Hits
	Motif LOGO	Motif_IUPAC_Code	*e*-Value	Nb of Sequences	Target_ID (*A. thaliana*)	*p*-Value	*e*-Value	*q*-Value
motif#1	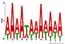	AAAATAWAWAWA	9.30 × 10^−79^	68	C2C2dof_tnt.At4g38000_col_a_m1	1,57 × 10^−5^	1,37 × 10^−2^	7,27 × 10^−3^
motif#2	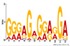	GGRAGAGRARGA	2.70 × 10^−45^	70	C2H2_tnt.TF3A_col_a_m1	5,28 × 10^−6^	4,61 × 10^−3^	9,20 × 10^−3^
motif#3	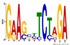	GAAGSYTCTAGA	3.90 × 10^−8^	49	HSF_tnt.HSFA1E_col_a_m1	2,79 × 10^−11^	2,43 × 10^−8^	4,76 × 10^−8^
motif#4	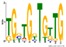	TGBTGKTGKTG	7.90 × 10^−8^	47	MYBrelated_tnt.AT1G18960_col_a_m1 (AT1G18960)	6,11 × 10^−5^	5,33 × 10^−2^	7,65 × 10^−2^

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
