# Peer review of "Dynamics of the Transcriptome Response to Heat in the Moss, Physcomitrella patens"

_ijms, 2020, doi:10.3390/ijms21041512_

Round 1
Reviewer 1 Report
Dear authors, thank you for resubmitting your manuscript and improving the quality of the presentation.
Overall, I found that the majority of the issues I had pointed out in the original version have been appropriately addressed. I still have some comments that could help to further improve the contents of the manuscript.
Section 2.1. – global analysis: The PCA clearly indicated that PC1 is explained by the “early response” factor at 1h, while PC2 is entirely explained by temperature. This should be pointed out and discussed here to add some further solidity to this analysis.
Figure 2 – there is a typo in the header of the “37x37” columns, as it should be “37x32”
Discussion – I think it should be explained why 24°C was selected as an “optimal” temperature. Many readers will not be familiar with the species, its habitat and geographical distribution. Also, about the discussion about the large difference in the response to 30 and 37°, it would be good to explain whether this species can face, in naturally occurring conditions, 30° or more for brief periods of time, i.e. provide clear limits of the natural range of temperature tolerance of the species.
Section 3.2: I would strongly suggest to decrease the size of this section and place it after section 3.3. After all, the majority of the genes responsive at 30° were also responsive at higher temperatures (37°), which suggests that a large component of the physiological response is shared. Such a response is probably just less intense at 30°, so the authors should try to discuss their data about 30° response in this manner (also taking into account that no 30°-specific GO tearms were identified!).
Section 4.3. Add some methodological details. Where have been the organisms maintained? With what photoperiod?
Section 4.5. “using one channel” does not mean anything. The authors probably mean “using one lane”. Replace “by paired-end” with “using a 2x100 bp paired-end strategy”. There is some redundant information at the end of the paragraph than can be removed.
Author Response
Reviewer 1
Dear authors, thank you for resubmitting your manuscript and improving the quality of the presentation.
Overall, I found that the majority of the issues I had pointed out in the original version have been appropriately addressed. I still have some comments that could help to further improve the contents of the manuscript.
Section 2.1. – global analysis: The PCA clearly indicated that PC1 is explained by the “early response” factor at 1h, while PC2 is entirely explained by temperature. This should be pointed out and discussed here to add some further solidity to this analysis.
Answer: We do agree with the reviewer #1. Sentences were added both in “Results” and “Discussion” sections.
Figure 2 – there is a typo in the header of the “37x37” columns, as it should be “37x32”
Answer: This was a mistake. It was corrected.
Discussion – I think it should be explained why 24°C was selected as an “optimal” temperature. Many readers will not be familiar with the species, its habitat and geographical distribution. Also, about the discussion about the large difference in the response to 30 and 37°, it would be good to explain whether this species can face, in naturally occurring conditions, 30° or more for brief periods of time, i.e. provide clear limits of the natural range of temperature tolerance of the species.
Answer: We added information at the beginning of “Discussion” section where we explained why we chose 30°C and 37°C.
Several studies have shown that P. patens is highly tolerant to dehydration, salinity, and other abiotic stress factors [references #51-54], and this tolerance is thought to be based on the mobilization of efficient defense and repair mechanisms in response to stress and during stress relief [reference #52]. Those studies used 24 °C as an optimal temperature for P. patens. Heat stress that has been investigated was a short heat stress (Chang et al., 2014 for example) followed by a period of recovery. However, this severe heat stress (37 °C) was not tested continuously. For the 30 °C treatment, we chose an intermediate temperature to address the question of adaptation to a moderate heat stress albeit physiologically less harmfull.
Saavedra L, Svensson J, Carballo V, Izmendi D, Welin B, Vidal S: A dehydrin gene in Physcomitrella patens is required for salt and osmotic stress tolerance. Plant J 2006, 45:237–249. Charron AJ, Quatrano RS: Between a rock and a dry place: the water-stressed moss. Mol Plant 2009, 2:478–486. Frank W, Ratnadewi D, Reski R: Physcomitrella patens is highly tolerant against drought, salt and osmotic stress. Planta 2005, 220:384–394. Wang XQ, Yang PF, Liu Z, Liu WZ, Hu Y, Chen H, Kuang TY, Pei ZM, Shen SH, He YK: Exploring the mechanism of Physcomitrella patens desiccation tolerance through a proteomic strategy. Plant Physiol 2009, 149:1739–1750.
“whether this species can face, in naturally occurring conditions, 30° or more for brief periods of time, i.e. provide clear limits of the natural range of temperature tolerance of the species.”: It is hard to answer this question because plants are continuously subjected to a combination of several environmental stimuli in nature. The outcome of this could be quite complex. For example, a temperature of 30°C is likely to be more damaging if it is accompanied with a water stress or with an excess of light.
Section 3.2: I would strongly suggest to decrease the size of this section and place it after section 3.3. After all, the majority of the genes responsive at 30° were also responsive at higher temperatures (37°), which suggests that a large component of the physiological response is shared. Such a response is probably just less intense at 30°, so the authors should try to discuss their data about 30° response in this manner (also taking into account that no 30°-specific GO tearms were identified!).
Answer: We changed the order but we thought it’s worth discussing the effect of a milder temperature than 37°C even though part of the heat response could be shared. Particularly, this section is mainly dedicated to the thiamine pathway involvement, which appears to be specific to this 30 °C-heat stress.
Section 4.3. Add some methodological details. Where have been the organisms maintained? With what photoperiod?
Answer: This information was given in the 4.1 section.
Section 4.5. “using one channel” does not mean anything. The authors probably mean “using one lane”. Replace “by paired-end” with “using a 2x100 bp paired-end strategy”. There is some redundant information at the end of the paragraph than can be removed.
Answer: This was corrected in the text.
Reviewer 2 Report
The revised manuscript was significantly improved, although I still suggest some editorial corrections listed below.
Introduction:
“Activated HSFs bind to promoters and initiate the transcription of heat responsive gene encoding, such as Heat Shock Proteins (HSPs)”.
It seems that some words are missing in this sentence.
“Besides the HSR basic mechanisms, it has been shown that alternative mRNA splicing is involved in the regulation of the HSR in P. Patens [21]”
Patens – small letter (the same comment is relevant for the Conclusions section)
Results:
“Figure 2: Venn diagram analysis at 37 °C and 30 °C”
The capture is very laconic. Please change it to something similar to: “Venn diagram of expression data analysis from 37 °C and 30 °C treatments…” – the same refers to other Figures with Venn diagrams.
Tables under Venn diagrams in Figure 2 have the headers “Biological process up (or down) 37 x 37”. Did you mean 37 and 30?
The last table – for down-regulated genes says: “No GO terms”. I think it is better to add corresponding header to this table and in the row below write “No significantly enriched GO terms”.
“The results indicated that, the yellow module was enriched in 6 GO terms, among which:”
Please change “among which” to “including”. The same comment is relevant to other similar expressions in the remaining part of this paragraph.
Moreover, the order of color-coded clusters are still not the same in the text, Figure 8 and in supplementary material. Such organization of the data makes it really hard to follow. Please use one uniform order to present this information.
Discussion:
“Those genes are mainly involved in protein folding, which may indicate that this metabolism is directly affected by the heat.”
I suggest to change “metabolism” to “process”
I also give my additional comment to one of authors’ answers:
A comment from first round review: Could you please give a short explanation in this section, why did you payed closer attention to yellow module of co-expressed genes and not to the other groups?
Answer: As a proof-of-concept, we choose the yellow module because it contained a majority of genes known to be rapidly activated by high temperature, such as those coding for HSPs. In this context, it was expected to find heat stress elements (HSE) within the promoters of these genes. Finding HSE could be considered as a proxy of the relevance of our approach. The cis-motifs within the promoters of co-expressed genes present in the other modules are gathered in supplemental data (Table S5).
I think you may include the above rationale in the text of your manuscript. It will allow the readers to follow your concept in an easier way.
To conclude, after the final (mostly editorial) modification of the manuscript I think it can be published in International Journal of Molecular Sciences.
Author Response
The revised manuscript was significantly improved, although I still suggest some editorial corrections listed below.
Introduction:
“Activated HSFs bind to promoters and initiate the transcription of heat responsive gene encoding, such as Heat Shock Proteins (HSPs)”.
It seems that some words are missing in this sentence.
Answer: This was corrected in the text.
“Besides the HSR basic mechanisms, it has been shown that alternative mRNA splicing is involved in the regulation of the HSR in P. Patens [21]”
Patens – small letter (the same comment is relevant for the Conclusions section)
Answer: This was corrected.
Results:
“Figure 2: Venn diagram analysis at 37 °C and 30 °C”
The capture is very laconic. Please change it to something similar to: “Venn diagram of expression data analysis from 37 °C and 30 °C treatments…” – the same refers to other Figures with Venn diagrams.
Answer: This was corrected in each figure presenting the Venn diagrams.
Tables under Venn diagrams in Figure 2 have the headers “Biological process up (or down) 37 x 37”. Did you mean 37 and 30?
Answer: This was a mistake. It was corrected.
The last table – for down-regulated genes says: “No GO terms”. I think it is better to add corresponding header to this table and in the row below write “No significantly enriched GO terms”.
Answer: We modified the figures 5 and 6 based on the comments.
“The results indicated that, the yellow module was enriched in 6 GO terms, among which:”
Please change “among which” to “including”. The same comment is relevant to other similar expressions in the remaining part of this paragraph.
Answer: This was changed in the whole paragraph.
Moreover, the order of color-coded clusters are still not the same in the text, Figure 8 and in supplementary material. Such organization of the data makes it really hard to follow. Please use one uniform order to present this information.
Answer: This was modified in order to respect the color-coded clusters in Table S3, Tale S4 and Table S5.
Discussion:
“Those genes are mainly involved in protein folding, which may indicate that this metabolism is directly affected by the heat.”
I suggest to change “metabolism” to “process”
Answer: This was modified.
I also give my additional comment to one of authors’ answers:
A comment from first round review: Could you please give a short explanation in this section, why did you payed closer attention to yellow module of co-expressed genes and not to the other groups?
Answer: As a proof-of-concept, we choose the yellow module because it contained a majority of genes known to be rapidly activated by high temperature, such as those coding for HSPs. In this context, it was expected to find heat stress elements (HSE) within the promoters of these genes. Finding HSE could be considered as a proxy of the relevance of our approach. The cis-motifs within the promoters of co-expressed genes present in the other modules are gathered in supplemental data (Table S5).
I think you may include the above rationale in the text of your manuscript. It will allow the readers to follow your concept in an easier way.
Answer: We agreed. We added the sentence at the beginning of the appropriate section.
To conclude, after the final (mostly editorial) modification of the manuscript I think it can be published in International Journal of Molecular Sciences.
Reviewer 3 Report
The authors have improved the manuscript following the reviewer requests, however,
I suggest the conclusion of the following answer within the manuscript:
"Answer: The experiments for this paper were performed on two cultivars; SxB49 and SxB139, of Triticum aestivum. These two Triticum aestivum L. recombinant inbred lines (RIL) are derived from a cross between two elite spring wheat varieties; Seri M82 and Babax (Pinto et al., Theor Appl Genet. 2010; https://doi.org/10.1007/s00122-010-1351-4). For more details, see Girousse et al. 2008."
Author Response
The authors have improved the manuscript following the reviewer requests, however,
I suggest the conclusion of the following answer within the manuscript:
"Answer: The experiments for this paper were performed on two cultivars; SxB49 and SxB139, of Triticum aestivum. These two Triticum aestivum L. recombinant inbred lines (RIL) are derived from a cross between two elite spring wheat varieties; Seri M82 and Babax (Pinto et al., Theor Appl Genet. 2010; https://doi.org/10.1007/s00122-010-1351-4). For more details, see Girousse et al. 2008."
Answer: We think that it is not the purpose of the manuscript and the citation of the reference should be enough in this paper.
Round 2
Reviewer 1 Report
All the remaining issues have been appropriately addressed.
This manuscript is a resubmission of an earlier submission. The following is a list of the peer review reports and author responses from that submission.
Round 1
Reviewer 1 Report
The manuscript by Elzanati and colleagues investigates the effect of mild and sever heat stress on Physcometrella patens through a transcriptomics approach.
I am sorry to say that this manuscript was quite hard to follow due to the very low quality of the English language used. It is absolutely mandatory that the authors carry out an extensive editing of the text, either involving a native English speaking colleague in the revision, or relying on a professional language editing service.
Overall, this was not a high quality work. Although the use of just two biological replicates is a rather important limitation of this study, the data generated could have been still analyzed in a meaningful way. However, the authors chose to use unreasonable thresholds for the detection of DEGs. Log2 FC >|2| seems to be way too low to take into account intraspecific variation, but my main concern is about the extremely permissive corrected p-value threshold (0.5). This was definitely unacceptable, taking into account the presence of just two biological replicates. As a rule of thumb, whenever few biological replicates are available, high stringency would be required.
Also, I am sorry to say that very scant information was provided about the methods used for mapping (which parameters were used? This is never mentioned…) and statistical analysis 8there is a link to ref. [43], which was not satisfying. The authors did no mention which library preparation kit was used, and many other crucial information is missing in the materials and methods section. For example, in section 4.9.1, the authors do not mention what expression values were used for this analysis. 4.9.2: explain why this was done. It does not seem right to re-annotate the genome only based on Arabidopsis.
“21 429 genes were identified whose level of expression was considered as significant and used for further analysis.” These criteria are not explained, so it is hard to understand what the authors are referring to.
Figure 2/3/4: these types of figure would be fine for a publication dedicated to a schoolbook, not for a scientific paper. Please replace these with tables with enrichment p-values instead.
In the materials and methods section, the authors very briefly mention that they carried out RT-PCR analysis, but apparently these data were not presented in the paper.
I will not go too much into detail about the many issues of this manuscript, but it looks like that the authors may have good data in their hands. The trends observed are consistent with expectations and the enriched GOs show potential interest. However, I see two big problems here: first, the manuscript is written with a very bad English language. Second, the methodological details provided are broadly insufficient and the authors never explained why some choices were made. Overall, the data was presented in an amateur-ish way, and it could be improved in a very significant way with an improved writing style.
I am encouraging the authors to deeply re-evaluate the writing of this manuscript and re-think the way the data is presented, aiming at specialists (tables would be much more informative than figures).
Reviewer 2 Report
The work presented by Elzanati et al. describes the response of P. patens transcriptome to two regimes of heat stress – mild (at 30 °C) and strong (37 °C), which was measured by RNA-Seq. Each of the stress regimes was applied for 1, 6, 12, and 24 hours and the stress-induced transcriptomic responses were compared to the same time points of plants grown in control conditions (at 24 °C). The paper gives an overview on the general differences in P. patens response to these stresses and points more attention to differentially expressed genes, which encode proteins involved in heat shock response. The paper, in my opinion, can be interesting for IJMS readers, but it requires several important corrections before its publication.
Starting from a general issue: after reading the results and discussion I am lacking a more clear distinction between information about control samples, mild heat treatment and the strong stress. At present these descriptions seem to overlap to some extent, what makes it very difficult to follow the author’s argumentation. Maybe dividing your results into sections dedicated to these two stress regimes will be better. It may also help to emphasize more the differences between the two stresses you applied.
Also, I am lacking a clearer narrative flow that may lead the reader throughout your manuscript – from general RNA-Seq data to the specific findings of the prevalence of heat shock response process after the application of stress, the importance of HSP-related genes in heat response and possible involvement of thiamine biosynthesis in the acclimation process. Please think, how to connect better all the information to lead the reader to the most important conclusions of your work. Also, indicate what is the most important novelty of your study.
Another thing is related to the dynamics of transcriptome changes over the time of stress application. You set this task as the objective of your study, but in the results this subject is not touched to the extend I expected. You mostly concentrated on the effect of 1 hour and 24 hours of treatments. It is interesting to find out why after 6 hours and 12 hours at 30 °C the number of DEGs drops down. Which genes remains to be differentially expressed at these time points? What is their function? Why most of genes up-regulated after the first hour of stress do not show differential expression later? Is there some connection here with the circadian clock, that you mentioned in your results? Do the DEGs discovered after 24 hours treatment at 30 °C encompass also the genes differentially expressed after the first hour of this stress? Or maybe at the prolonged time you observe different genes than at the beginning? And how this dynamics of transcriptome changes at 30 °C corresponds to the differences observed for the more severe stress conditions? I am lacking a more comprehensive description in this area.
And although I am not a native English speaker, I think that some (minor) English editing is also necessary for the manuscript.
More specific remarks and questions are as follows.
Page 3, lines 18-19: “Interestingly, the 1-hour timepoint is clearly distinguishable from the remaining timepoints indicating the influence of the circadian clock”.
This is true for control samples, but not for the stressed ones. You write about it in the next sentences, but starting from such general statement may confuse the reader. It should be clearly stated at the beginning that such difference refers to the controls only.
Moreover, the PCA plot shows that the stress may possibly have an impact on the circadian clock in P. patens. Having the RNA-Seq data from control samples at different time points, is it possible to extract more information on the genes possibly related to the circadian clock and see if (or how) their expression is affected by the stress? There are some data in the literature showing that also HSF proteins are important to maintain the circadian rhythm under temperature stress (for example reviewed in: Gil KE, Park CM. 2019. Thermal adaptation and plasticity of the plant circadian clock. New Phytol. 221(3):1215-1229. doi: 10.1111/nph.15518).
Page 3, lines 21-22: “Therefore, comparison between heat-treated samples and controls will be made at each time point”.
I think past tense is better in this sentence. Moreover, it is not clear what do you mean by “each time point”. I understand that you compared samples treated by the stress at specific time (for example for 1 hour) to the control samples grown for the same period of time (1 hour), but it does not sound in this way from the above sentence.
Page 4, lines 12-14. This part refers to the description of data in Table 1, but both – text and the Table do not correspond to each other. In the text you summarize the number of all DEGs, whereas in the Table 1 are numbers for specific comparisons between control and stressed samples. These numbers obviously shows some redundancy, as part of DEGs from each of the comparisons will overlap with the other. It may be better to show such results as Venn diagrams to present the overlapping groups of DEGs.
Also in the line 14 you wrote: “only 62 genes are both repressed and activated”. Such number cannot be extracted from Table 1. Besides, a gene cannot be repressed and activated in the same time – this sentence needs rewording.
Section 2.3. Ontology of DEGs
Here I have another general comment on listing the most significant GO terms. I think it is not necessary to give a series of GO numbers in the main text, as it disrupts the reading very much. Their description should be enough, with FDR values in the brackets, to show their significance.
Page 5 line 9: “No significant GO term was found at 6 h or 12 h”.
I understand that the GO enrichment analysis did not give any results here, due to the limited number of DEGs, but later in the discussion you refer to these genes, and describe their ontology (Page 11, lines 3-5). I think it is important to add such functional information also to the results.
Page 5, lines 19-20: “Regarding the down-regulated genes, pretty early after heat treatment (6 h), it seemed that the defense mechanisms, as well as the cell cycle (Table S2), were repressed”.
In table S2 several GO ontologies for negative regulation of cell division are presented. If a negative regulator of some process is repressed it means that this process may actually be activated. Please verify the above statement.
Moreover, I could not find any data on GO enrichment for DEGs downregulated after 1 hour treatment at 37 °C.
Page 6 line 8 and the following.
At the beginning of this section I am missing an information that clearly says what kind of expression profiles are “hidden” under your color-coded modules. Moreover, the order in which you mentioned the color-coded modules in the text, on the Figures 5 and 6 and also in Supplementary Tables S4 and S5 is every time different. It makes difficult to follow these data in your manuscript. Also, is it necessary to have two separate figures (5 and 6) for all of the co-expression profiles?
Section 2.4. Conserved motifs within promoters of correlated genes.
Could you please give a short explanation in this section, why did you payed closer attention to yellow module of co-expressed genes and not to the other groups?
I am also missing an argumentation why did you decided to concentrate on genes with a motif recognized by HSF proteins, and not on the genes with others motives in their promoters.
Page 11, lines 11-14: “It therefore seems that very soon after the application of the temperature of 30 °C (i.e. from 1 h), there is a response of the “stress response” type involving HSP type genes, then it there would be an acclimation phase which is relayed at 24 h by the activation of thiamine synthesis”.
What do you mean by “relayed”?
Page 23, lines 23-38.
Here you discuss the possible importance of thiamine biosynthesis process in heat response.
I think it is good to add to the Results an information that after 24 hours of 30 °C treatment the thiamine biosynthesis (and related thiazole metabolism) were the only significant GO categories that emerged from your data. One can read it from Supplementary Table, but having such information in the main text will add a better link between your results and the discussion.
Page 12 lines 34-36: “Besides the up-regulated genes, we found that the number of down-regulated ones steadily increased with the duration of heat stress. This variation could be interpreted as merely a direct consequence of the damaging effects of heat stress on protein stability”.
Is this statement legitimate? Why loss of protein stability should lead to down-regulation of its gene expression? Is it possible to draw such link between protein unfolding and down-regulation of transcription?
Page 13 lines 3-5: “This finding suggests that the crosstalk between activation of HSR and repression of PCD might be a general molecular signaling mechanism in land plants”.
Again I am not sure if such statement is legitimate. Simultaneous observation of two events such as up-regulation of one group of genes and down-regulation of other group is not enough to say that there is a crosstalk between such groups. Do you have other data to support such conclusion?
Page 13, lines 25-26: “Indeed, Maruyama et al. [39] showed that the heat stress element (HSE) should be included in a particular 18 nucleotide sequence to yield a heat stress responsiveness in some crop plants”.
Did the HSE motives in your DEGs occur in a similar context of such 18 nucleotide sequence as discovered by Maruyama et al.?
Page 14, line 11.
Here you write about qRT-PCR method, but in the context of cDNA synthesis only (cDNA synthesis is not a qRT-PCR). Did you also made qPCR analysis of some of your genes?
Section 4.9. Identification of differentially expressed genes (DEG)
Here you should also write how did you compare your treated samples to the control taking into account the time points used.
Figures 2, 3 and 4 – I think that presenting the GO enrichment using Wordle tool may be interesting for conference presentation, but in my (subjective) opinion, it does not have the scientific “sound” for presentation in the manuscript. Moreover, such figure may not be readable, especially in the case of smaller GO categories. I suggest, you may consider a different way of presentation of these data (maybe histogram based on GO slim analysis).
Figure S1 – I cannot find this figure in the supplementary materials. Please verify if you uploaded this figure together with other files.
Reviewer 3 Report
Elzanati et al. investigated the effects of mild (30 °C) and high (37 °C) continuous heat stresses up to 24 h on the remodeling of the transcriptome in the moss Physcomitrella patens using RNAseq.
The manuscript is well written and easy to read
The authors should report the corresponding scientific name of the crop cited in the manuscript
The authors should perform the statistical analysis (e.g. MANOVA) to confirm the differences between the reported clusters
Page 11 line 20. On wheat, (bred or durum?), which cultivar?
Page 11 line 31. On maize, which cultivar?
When the authors compare the obtained results with results available in the literature, comparing data obtained on other crop should be reported the name of the cultivar because, different cultivars have different genetic background influencing the resistance/tolerance to different abiotic and biotic stresses.
Conclusions are too general, please improve this section